# Toxicity and Starvation Induce Major Trophic Isotope Variation in *Daphnia* Individuals: A Diet Switch Experiment Using Eight Phytoplankton Species of Differing Nutritional Quality

**DOI:** 10.3390/biology11121816

**Published:** 2022-12-14

**Authors:** Michelle Helmer, Desiree Helmer, Dominik Martin-Creuzburg, Karl-Otto Rothhaupt, Elizabeth Yohannes

**Affiliations:** 1Limnological Institute, University of Konstanz, Mainaustrasse 252, 78464 Konstanz, Germany; 2Research Station Bad Saarow, Department of Aquatic Ecology, BTU Cottbus-Senftenberg, Seestrasse 45, 15526 Bad Saarow, Germany

**Keywords:** stable isotopes, cyanobacteria, turnover-rates, discrimination factor, *Planktothrix*, *δ*^13^C, *δ*^15^N

## Abstract

**Simple Summary:**

The estimates of animal diets and trophic structures using stable isotope analyses are highly influenced by the diet–tissue discrimination and tissue turnover rates. However, these factors are often unknown because they must be measured using controlled feeding studies. Furthermore, these parameters may be influenced by the diet quality, quantity, toxic stress, and starvation or fasting, as well as other factors. We measured the effects of toxic stress, starvation, and diet quality on the turnover rate and diet–tissue discrimination in *Daphnia* individuals. We raised individuals with a common laboratory diet and switched them to eight different dietary sources with varying levels of nutritional quality, while one group experienced starvation. The isotopic values were assessed on a daily basis post-diet change. Overall, we showed that in addition to the nutritional quality, toxic stress and starvation are the main processes that affect the two key parameters of the stable isotope analysis.

**Abstract:**

Stable isotope values can express resource usage by organisms, but their precise interpretation is predicated using a controlled experiment-based validation process. Here, we develop a stable isotope tracking approach towards exploring resource shifts in a key primary consumer species *Daphnia magna*. We used a diet switch experiment and model fitting to quantify the stable carbon (*δ*
^13^C) and nitrogen (*δ*
^15^N) isotope turnover rates and discrimination factors for eight dietary sources of the plankton species that differ in their cellular organization (unicellular or filamentous), pigment and nutrient compositions (sterols and polyunsaturated fatty acids), and secondary metabolite production rates. We also conduct a starvation experiment. We evaluate nine tissue turnover models using Akaike’s information criterion and estimate the repetitive trophic discrimination factors. Using the parameter estimates, we calculate the hourly stable isotope turnover rates. We report an exceedingly faster turnover value following dietary switching (72 to 96 h) and a measurable variation in trophic discrimination factors. The results show that toxic stress and the dietary quantity and quality induce trophic isotope variation in *Daphnia* individuals. This study provides insight into the physiological processes that underpin stable isotope patterns. We explicitly test multiple alternative dietary sources and fasting and discuss the parameters that are fundamental for field- and laboratory-based stable isotope studies.

## 1. Introduction

Studies on trophic structures and community dynamics are central towards understanding community ecology [1,2]. Trophic structures are based on food web networks linked by feeding interactions, which ultimately define the ecosystem structure [3]. Thus far, much work on the spatial and temporal dynamic nature of trophic structures has been conducted, and it is now well recognized that trophic variation occurs among and within ecosystems across multiple scales [4]. In an aquatic ecosystem, energy is transferred from the primary consumer level to higher levels, and a balance in these transfers is vital to the health and stability of an ecosystem [5]. However, knowledge gaps remain for most baseline primary consumers, such as *Daphnia*, which cannot be estimated using evolving technological tracking tools for a more comprehensive, ecosystem-based understanding (such as for those in fish). A stable isotope analysis (SIA) is a useful tool that has rapidly expanded in primary consumer aquatic ecosystem studies. Such studies using SIAs rely on the fact that consumers reflect the isotopic composition of the consumed prey item, which varies with prey ecology, geographic location, and habitat conditions [6]. This fundament has allowed investigations on consumer diets, trophic dynamics, and habitat use with SIAs, most often by applying tissue carbon (*δ*
^13^C) and nitrogen (*δ*
^15^N).

Understanding isotopic turnover rates in consumer tissues requires the quantification of the temporal scale of diet shifts. For instance, following a shift towards an isotopically varying dietary source (e.g., algae), the *δ*
^13^C and *δ*
^15^N values of primary consumer tissues (e.g., *Daphnia*) change over a period of time until arriving at a consistent value (steady-state conditions) that mirrors the new dietary isotope value. The rate of change is driven by the physiological processes of tissue synthesis (e.g., growth) and anabolic and catabolic processes (e.g., fatty acid metabolism, detoxification). Briefly, the isotopic compositions of primary consumer tissues reflect those of the diet.

In recent years, dramatic shifts in global and local environmental conditions have been observed. Such shifts are often driven by climate change, with major impacts on alterations in temperature [7], elevated atmospheric CO_2_ [8,9], and nutrient availability in ecosystems [10]. Additionally, humans have a direct influence on the ecosystem nutrient dynamics through supplementary feeding and fertilizer application [11,12,13]. In the context of lake productivity, such ongoing changes result in changes in the physico- and bio-chemical properties of lakes, hampering the growth of eukaryotic algae, e.g., *Chlorella* sp. [14,15] and *Scenedesmus* [16], but favoring cyanobacteria [17,18,19,20,21,22,23,24]. Some species (e.g., of the genus *Microcystis*) are directly favored by warmer surface water temperatures coupled to eutrophying nutrient inputs [17,24,25,26,27,28], while others (e.g., *Planktothrix rubescens*) are associated with stronger thermal stability of the water column, which may generate a rapid shift in the quality or quantity of nutrient sources [29,30,31] in the eutrophic zone. On the other hand, filamentous, diazotrophic cyanobacteria of the genus *Dolichospermum* are among the most ubiquitous bloom-forming cyanobacteria, and their dominance and persistence have increased due to anthropogenic eutrophication and global climate change [32]. As a result, it is expected that herbivorous zooplankters will be increasingly confronted with unicellular, colony-forming, and filamentous cyanobacteria in the future.

Controlled laboratory-based experiments quantifying primary producer and consumer-specific isotopic turnover and discrimination factors may improve the knowledge on stable isotope dynamics and its direct use towards the field study of stable isotope (SI) ecology.

Stable isotope (SI) turnover rates permit measurements of the temporal scale of the diet through proxy demonstrations of the dietary source represented by the tissue SIA composition. Moreover, the turnover rates can be applied as an ‘isotopic clock’ approach (e.g., Madigan, et al. [33]), using isotopic endmembers, such as diet or aquatic water depth and consumer SIA values, to measure the temporal scale of diet shifts, habitat changes, and movements. Nonetheless, the accuracy of these spatial frames and timeframes are maximized with species-specific isotopic turnover rates and discrimination factors, which are best calculated from laboratory experiments. Cladocerans of the genus Daphnia are key species in many lentic habitats that exhibit a variety of adaptive responses to rapid environmental fluctuations [34] and are representative as baseline organisms in freshwater food webs towards understanding the SIA ecology at the herbivore–grazer interface. However, the lack of experimentally derived isotope turnover parameters and discrimination factors for Daphnia limits the interpretation of SIA data for (a) spatial and temporal patterns in baseline food web and higher trophic-level dynamics and (b) temporal scales of movement patterns within lake columns (e.g., pelagic vs. benthic and pelagic vs. littoral).

The aim of the study was to evaluate whole-body isotopic turnover rates and diet–tissue discrimination factors in individual Daphnia magna. The results of the study (1) could be applied to field-collected SIA data for active, baseline trophic-level studies in freshwater systems and (2) could help to improve predictions of the responses of Daphnia isotopic values towards cyanobacterial dynamics that could develop in times of climate change.

## 2. Materials and Methods

### 2.1. Phytoplankton Culturing and Preparation

As stock cultures of *Daphnia magna* we used the green alga *Acutodesmus obliquus* (SAG 276-3a) as food, which was cultured in Cyano medium [35] in 5 L batch cultures under permanent illumination (24 h light). The food was harvested in the late-exponential growth period.

We considered a range of different phytoplankton species that are known and expected to be potential dietary sources for herbivorous zooplankters. Specifically, we used eight different algae species (Table 1) for the diet switch experiment. The phytoplankton species differed in their cellular organization (unicellular or filamentous), photopigment and biochemical compositions (sterols and polyunsaturated fatty acids (PUFA)), as well as in the production of secondary metabolites and stable isotopic signatures (Table 2). Each phytoplankton species was cultured semicontinuously in modified Woods Hole (WC) medium without vitamins [36] at 20 °C with illumination at 62 × 10^15^ mol quanta m^−2^ s^−1^ and a light/dark cycle of 16:8. The algae were harvested in the late-exponential growth phase. The carbon concentrations of the different food suspensions were estimated from photometric light extinction (480 nm) and carbon extinction equations determined prior to the experiment.

To determine the stable isotopic signature of each phytoplankton species, samples of each food suspension were taken every day and filtered onto pre-annealed GF/F filters (Whatman™, GE Healthcare Life Science, Chicago, IL, USA) over the whole experimental period (four days). Subsequently, the filters were dried at 50 °C and stored in a desiccator until further analysis.

### 2.2. Diet Switch Experimental Setup

The diet switch experiment was conducted with third-clutch juveniles (born within <12 h) of *D. magna* clone S5 (originally isolated in Sheffield). The *D. magna* samples were kept in glass beakers filled with 1.4 L of filtered lake water (0.2 µm pore-sized membrane filter) at 20 °C and with a light/dark cycle of 18:6, with 20 individuals per beaker. Every other day over a period of seven days, the animals were transferred to new beakers containing freshly prepared food (2 mg C L^−1^
*A. obliquus*).

After seven days of growth, all *Daphnia* samples were subject to a simultaneous diet switch. The animals were transferred to glass beakers filled with 200 mL of filtered lake water containing 2 mg C L^−1^ each of the different algae. Each treatment consisted of six replicates per time point with five *Daphnia* samples per beaker. The animals were transferred every day to new beakers with freshly prepared food suspensions for the different treatments.

After 0 h, 24 h, 48 h, 72 h, and 96 h, six beakers (replicates) were subsampled from each treatment. The animals were washed three times with demineralized water, transferred into tin cups, dried for 24 h, weighed on an electronic balance, and stored in a desiccator until further analysis.

### 2.3. Stable Isotope Analysis

The *Daphnia* samples were dried and 0.3–0.7 mg was weighted in small tin cups to the nearest 0.0001 mg, using a microanalytical balance (Sartorius 4504MP8). The filters of the algae were also dried and packed (1.5–2 mg) into small tin cups.

The stable isotope analyses were conducted in the stable isotope laboratory of the Limnological Institute, University of Konstanz (Konstanz, Germany). The zooplankton and phytoplankton samples were combusted in a Vario Micro-Cube elemental analyzer (Elementar Analysensysteme GmbH, Hanau, Germany) connected to an isotope ratio mass spectrometer (Isoprime Ltd., Cheadle Hulme, UK) for the determination of ^13^C/^12^C and ^15^N/^14^N. The accuracy of the instrument was assessed by measuring internal standards along with samples. The stable isotope data are reported using the *δ*-notations (*δ*
^13^C and *δ*
^15^N) in parts per thousand (‰), where:(1)$$\delta =1000\times \left(\frac{R_{sample}}{R_{standard}}\right)-1$$

The data were relative to the Peedee Belemnite (PDB) standard for carbon and atmospheric dinitrogen (N_2_) for nitrogen. Two casein samples were placed between eight unknowns in sequence as a laboratory standard.

Briefly, a total of 111 *Daphnia* samples (3 replicates of each treatment and time point) and 38 phytoplankton samples (a single replicate of each time point) were used for the stable isotope analysis. The *δ*
^13^C and *δ*
^15^N analyses were conducted for all samples.

### 2.4. Time-Based Stable Isotope Turnover Rates

The time-based turnover rates were calculated for the changes in *δ*
^13^C and *δ*
^15^N as an exponential function of time following the diet switch using the model of Hobson and Clark [56]:(2)$$\delta t=\delta_{eq}+\left(\delta_{0}-\delta_{eq}\right)e^{-\lambda t}$$

Here:

δt represents the *δ*
^13^C and *δ*
^15^N values of *D. magna* at the experimental time *t*;

$\delta_{eq}$ is the calculated asymptotic equilibrium with the new diet;

$\delta_{0}$ is the initial isotopic value prior to the diet switch;

λ is the turnover rate (h^−1^).

For the estimation of the variables in this model, we used the nls function in *R* with the self-starting asymptotic regression model SSasymp with the following equation:(3)$$\delta t=\delta_{eq}+\left(\delta_{0}-\delta_{eq}\right)e^{-\exp\left(\log_{\lambda }\right)t}$$

The turnover rate expressed in terms of the half-life (*T*_0.5_), the time period needed to achieve a 50% turnover of the isotopic composition of *δ*
^13^C, was calculated as [56]:(4)$$T_{50}=\frac{\ln\left(2\right)}{\lambda }$$

### 2.5. Determination of the Best Model Fit Using AIC

We also calculated Akaike’s information criterion for samples size scores (*AIC*) to evaluate the relative support for each model and to determine how well the exponential or linear model provided a better fit for the data:(5)$$\delta t=\delta_{eq}-\lambda \left(\delta_{0}-\delta_{eq}\right)t$$

The model assumptions were checked using nlstools package and check of normal distribution of Residuals using Shapiro Wilks test. The *AIC* differences (Δ*AIC*) between the two different models (exponential and linear) were calculated as
(6)$$\Delta AIC=AIC_{m}-AIC_{n}$$

Here:

$AIC_{m}$ is the calculated *AIC* of a model;

$AIC_{n}$ is the lowest *AIC* of the competing model.

### 2.6. Discrimination Factor (DF)

The isotopic differences between animal tissue and their diets during the experiment were estimated by subtracting the animal isotope values from those of their respective diets for two elements (*δ*
^13^C and *δ*
^15^N):(7)$$\Delta_{\left(tissue-diet\right)}=\delta_{tissue}-\delta_{diet}$$

### 2.7. Statistical Analysis

All statistical analyses were performed using R (version 3.6.2). The data were checked for the normality (Shapiro–Wilk Test) and homogeneity of variances (Levene test). For comparison of the *δ*
^13^C signatures of the diet and the *D. magna* sample from the last timepoint (*M. aeruginosa* 72 h, all other dietary treatments at 96 h after diet changed), we used the parametric paired t-test. Due to the non-normal distribution of the pooled data, we used the non-parametric Wilcoxon test.

To compare the difference between the isotopic signatures of *D. magna* and the diet after 96 h of incubation, depending on normal-distribution, we used the paired t-test and the Wilcoxon test. For the analysis of the differences between the stable isotopic signatures of the respective diets over time, we used a homogeneity of variance ANOVA and a post hoc Tukey test.

A cluster analysis was performed using the R packages factoextra, ggpubr, and cluster. To define the optimal number of clusters, we used the average silhouette method, the elbow method, as well as the gap statistic method [57].

## 3. Results

### 3.1. Isotopic Changes in Daphnia Tissues (δ ^13^C and δ ^15^N) after Diet Switch

Due to the increasing mortality rate of the *Daphnia* samples with the dietary treatment of *M. aeruginosa*, no animals survived 96 h after the diet switch (Figure 1). Therefore, only data up to 72 h are shown in Figure 1 and were used for the statistical analysis. Compared to this, *D. magna* with the other dietary treatments had a 100% survival rate.

Figure 2 and Figure 3 exhibit the *δ*
^13^C and *δ*
^15^N values for *D. magna* with the different dietary treatments and starvation over a time period of 96 h, respectively. Appendix A shows the SI values for *D. magna* with the different treatments listed.

In the following, the results for the various algal dietary sources are depicted, and all values are given as means ± SE.

#### 3.1.1. Chlorophytes: *C. klinobasis*, *C. vulgaris*, and *A. obliquus*

*D. magna* showed an initial rapid change in *δ*
^13^C when fed on the chlorophytes *C. klinobasis, C. vulgaris,* and *A. obliquus* (Figure 2). The rate of incorporation for algal carbon reached an equilibrium after 72–96 h. While the *δ*
^13^C values of *D. magna* changed rapidly (−4.92‰ ± 0.26), the *δ*
^15^N values remained almost constant (−0.17‰ ± 0.74) (Figure 3).

#### 3.1.2. Non-Toxic Cyanobacteria: *S. elongatus* and *T. variabilis*

Compared to the green algal carbon values, cyanobacterial carbon represented by *S. elongatus* (Bo8801 and Bo8809) and *T. variabilis* was incorporated rather slowly (within 48 h). After 48 h, the *Daphnia* did not incorporate the cyanobacterial carbon anymore and reached its equilibrium, while the *δ*
^15^N values slightly increased (0.9‰ ± 0.82) (Figure 2 and Figure 3).

#### 3.1.3. Toxic Filamentous Cyanobacteria: *P. rubescens* and *P. agardhii*

When fed with the filamentous cyanobacteria *P. rubescens* and *P. agardhii*, which do not contain sterols or long-chain PUFAs, and in addition produce a number of harmful secondary metabolites (Table 1), the mean (± SE) *δ*
^13^C values exhibited a minor shift (*P. rubescens*: = −0.27‰ ± 0.26; *P. agardhii:* 0.74‰ ± 0.16), while the *δ*
^15^N values increased after 48–72 h rather rapidly (*P. rubescens:* 5.15‰ ± 0.85; *P. agardhii:* 1.98‰ ± 0.59) (Figure 2 and Figure 3).

#### 3.1.4. Toxic Unicellular Cyanobacteria and Starvation

The *Daphnia* samples that were fed on toxic *Microcystis aeruginosa* increased in *δ*
^13^C (1.57‰ ± 0.25‰) as well as in *δ*
^15^N (1.17‰ ± 0.59) compared to the initial values (Figure 4).

The starving *D. magna* samples showed a different SI value compared to all the other dietary treatments. While the *δ*
^13^C signature of *D. magna* increased (1.17‰ ± 0.59) comparable to the SI values of *D. magna* with the toxic cyanobacterial diet, the *δ*
^15^N values decreased (−0.32‰ ± 0.51‰), as did the *D. magna* signature of the green algal diet compared to their initial treatment. Even 96 h after the dietary switch, *D. magna* did not reach equilibrium in terms of *δ*
^13^C.

Briefly, we note that the highest enrichment for *δ*
^13^C after 72 h in the *D. magna* tissue was for the dietary treatment with *M. aeruginosa.* The overall *δ*
^13^C value of *D. magna* held with the *M. aeruginosa* diet treatment was 1.57‰ (±0.25), which was enriched compared to the initial value, while the *δ*
^13^C value of the starving *D. magna* tissue was enriched by 1.17‰ (±0.59) after 96 h. On the other hand, the animals fed with *C. vulgaris* showed the steepest slope, with a decrease in the *δ*
^13^C signature of 5.00‰ (±0.38) compared to their initial signature (Figure 4).

The highest enrichment of *δ*
^15^N for the *D. magna* tissue after 96 h compared to the start signature was found in *P. rubescens* with 4.15‰ (±0.85), followed by the dietary treatment with *P. agardhii* with 1.98‰ (±0.59) (Figure 4). The largest decrease in the *δ*
^15^N signature of *D. magna* compared to the initial signature was found in *C. klinobasis* with −0.83‰ (±0.56), followed by the starvation treatment with a decrease of −0.32‰ (±0.51) (Figure 4).

### 3.2. Trophic Discrimination Factors (TDFs)

The trophic discrimination factors are shown in Figure 5 and listed in Appendix A.

#### 3.2.1. TDF: Chlorophytes

The *δ*
^13^C value of *D. magna* individuals fed on chlorophytes was enriched by 0.84‰ (±0.49‰), while the *δ*
^15^N value increased by 5.47‰ (±0.26‰).

Specifically, the *δ*
^13^C value of *D. magna* with the *C. vulgaris* dietary treatment did not show a significant difference from the dietary source (*p* > 0.05) (Table 3), while the *δ*
^13^C signatures of all other *D. magna* samples differed significantly (*p* < 0.05) from their diet source (Figure 5).

#### 3.2.2. TDF: Non-Toxic Cyanobacteria

The *D. magna* samples feeding on cyanobacteria of both strains of *S. elongatus* and of *T. variabilis* were 2.17‰ (±0.44‰) enriched in *δ*
^13^C and 5.70‰ (±0.68‰) enriched in *δ*
^15^N. Their *δ*
^13^C values also differed significantly (*p* < 0.05) from those of their diet. During the 96 h following the dietary switch, the Δ^15^N values increased by about 0.976‰ from 4.73‰ to 5.70‰, while the Δ^13^C values decreased by about 1.95‰ from 4.12‰ to 2.17‰.

#### 3.2.3. TDF: Toxic Cyanobacterial

The *D. magna* samples on the treatments with *M. aeruginosa, P. rubescens,* and *P. agardhii* did not incorporate the *δ*
^13^C signature of their diet. While the Δ^13^C values of *D. magna* samples with the other treatments decreased, the Δ^13^C value of these *D. magna* samples increased during the experiment by about 0.53‰ from 6.53‰ to 7.06‰. Additionally, the Δ^15^N value increased rapidly about 4.36‰ from 2.64‰ to 7.00‰. The highest enrichment compared to the dietary source was found in *P. rubescens,* with an average increase of 7.64‰ (±1.12), followed by *P. agardhii,* with an average increase of 6.36‰ (±0.40).

### 3.3. Cluster Analysis

A cluster analysis based on the *δ*
^13^C as well as the Δ^13^C values of *D. magna* tissues with the different dietary treatments (Figure 6) identified 4 distinctive groups. Cluster I was formed by the non-toxic dietary treatment with *T. variabilis* as well as both strains of *S. elongatus.* Cluster II represents the toxic dietary treatments with *P. agardhii, P rubescens,* and *M. aeruginosa,* while cluster III was formed by the chlorophytes *A. obliquus, C. klinobasis,* and *C. vulgaris*. Cluster IV was represented by starving *D. magna* samples. Based on the cluster analysis, the data for the different treatments were pooled for better fitting of the decay models.

### 3.4. Turnover Rates and Decay Models

The results of the comparison of Akaike’s information criteria (∆AICc) and the calculated parameters of the decay function are shown in Table 4.

The best fit for the *δ*
^13^C values of *T. variabilis, S. elongatus* (green), and *S. elongatus* (red) (difference in AICc: 14.21), as well as for *A. obliquus, C. klinobasis,* and *Chlorella* sp. (difference in AICc: 7.37), correspond to the one-phase exponential decay. Due to the increasing or stagnating *δ*
^13^C values of *M. aeruginosa,* the *P. rubescens* and *P. agardhii* turnover rates and decay models could not be calculated.

The calculated carbon turnover rates (*T*_0.5_) for *D. magna* fed on chlorophytes were very similar to each other and differed only slightly (*T*_0.5_: 35.52 h ± 1.49). The carbon turnover rates of the cyanobacterial strains *S. elongatus* and *T. variabilis* were compared to the chlorophytes, being 7.89 h faster. The average turnover time was 27.63 h, with the fastest turnover being for the red strain of *S. elongatus* at 23.78 h and the slowest turnover rate being for *T. variabilis* at 55.69 h.

## 4. Discussion

The results of these controlled laboratory experiments using eight species of algae fed to *Daphnia* samples demonstrated that the dietary quantity and quality as well as the toxic stress (presumably through production of harmful secondary metabolites) exert a measurable influence on the fractionation and incorporation rates of the carbon and nitrogen isotope values of the key aquatic consumer *Daphnia*.

### 4.1. Starvation

The starving animals in the present study showed enriched *δ*
^13^C values (1.17‰ ± 0.59), while the *δ*
^15^N values decreased by approximately 0.32‰ (±0.51) over a time period of 96 h. A similar effect of starvation on the *δ*
^13^C values had been documented earlier by Webb et al. [58] and Oelbermann and Scheu [59]. However, the decrease in *δ*
^15^N was contrary to the result a growing number of studies have shown, whereby nutritional stress (starvation) leads to an enrichment of the *δ*
^15^N values in consumer tissues in invertebrates [59,60,61].

In fact, Doi et al., 2017 conducted a meta-analysis of the *δ*
^13^C and *δ*
^15^N values of consumers post- and pre-starvation and showed a large variation in consumer isotope values (*δ*
^13^C range: −1.92 to 2.62‰; *δ*
^15^N range: −0.82 to 4.30‰). The analysis also showed both increases and decreases in *δ* ^13^C due to starvation, while the *δ*
^15^N values of most consumers increased along the length of the starvation period. Our findings also suggest that starvation induces changes in consumer *δ*
^15^N values, which are mainly explained by the length of the fasting period. Previously, Adams and Sterner [61] showed that starvation in *Daphnia magna* leads to *δ*
^15^N enrichment. While the available nitrogen decreases, the organisms are forced to recycle existing internal nitrogen reserves, which results in increasing *δ*
^15^N values as the lean body mass is lost without the replacement of excreted ^14^N [62].

There is a wide variety of reports on the effects of starvation on consumer stable isotope values. While some studies report *δ*
^15^N enrichments, no alteration [60], or enriched ^13^C [59] due to starvation, other studies have shown *δ*
^13^C enrichment [63] or no effect [64]. Elsewhere, chironomid starvation resulted in no change in *δ*
^15^N values but a significant enrichment in *δ*
^13^C values, presumably due to the preferential degradation of low-*δ*
^13^C components during periods of starvation [63]. In contrast to the other findings, Gorokhova and Hansson [64] could not observe any effect of starvation on the stable isotopic composition of mysids. The process of nutrient recycling and trophic enrichment due to starvation might induce a complex process and very variable isotope values within different species, life stages, and body conditions.

### 4.2. Chlorophytes

In contrast to starving animals, we observed a rapid change in the isotopic signature in *Daphnia* tissues feeding on chlorophytes. They reached isotopic equilibrium with their diet after 72–96 h following the diet switch. After 96 h, the *D. magna* tissue showed a slight enrichment of 0.84‰ in Δ^13^C and enriched Δ^15^N values (Δ^15^N 5.47). Although there was a slight decrease and an increase for the two isotopes, this result was in agreement with previous values obtained from 25 different lakes lying in temperate zones with average fractionation values of *δ*
^13^C ca. 0.4‰ (±1.3‰) and *δ*
^15^N 3.4‰ (±1‰). Earlier, Minagawa and Wada [65] documented Δ^15^N enrichment rates for all consumers (zooplankton, fish, and birds) of 1.3 to 5.3 (averaging 3.4‰ ± 1.1‰) with increasing trophic levels.

There were only slightly differences in isotopic fractionation between the different chlorophytes used as dietary sources. While the *D. magna* tissue fed on *Chlorella vulgaris* did not differ after 96 h from that of their diet, the isotopic signature of *D. magna* samples fed on *C. klinobasis* and *A. obliquus* still differed significantly from their dietary source after 96 h. Presumably, the biochemical composition of the different chlorophyte species might result in differences in the isotopic composition of *Daphnia* fed on a specific diet. Indeed, *Chlorella* species are known for the high variability of their biochemical composition [66]. Studies on the algal species *Chlorella vulgaris* showed that in addition to phytosterols, which *Daphnia* can convert to the required or important cholesterol, this alga also contains long-chain polyunsaturated fatty acids (PUFAs) such as ALA, EPA, and DHA [55,67,68,69,70], which are important for *Daphnia*. Evidently, the availability of sterols influences the somatic growth of *Daphnia*, while the PUFAs primarily play a crucial role in *Daphnia* reproduction [71]. Due to the essential nutrients present, *Chlorella vulgaris* can be considered a suitable food for *Daphnia*.

In aquatic food webs, the lipid composition of the prey is of great importance for the efficiency of the transfer of energy to higher trophic levels. In addition to the lipid composition, the elemental C/N ratio of the diet plays a decisive role in the isotopic fractionation in *Daphnia* and especially in the *δ*
^15^N enrichment of consumers. Adams and Sterner [61] found an inversely relationship between the *δ*
^15^N values of *Daphnia* and the diet–tissue isotopic fractionation factor with the nitrogen content of the phytoplankton’s diet. In addition to the biochemical composition of the phytoplankton, the variability of the C/N ratio could also cause an altered isotopic fractionation in *Daphnia*.

### 4.3. Non-Toxic Cyanobacteria

The traits of the cyanobacteria, such as the absence of long-chain PUFAs [72] and sterols [73], morphological properties (filamentous, pico-sized, colony-forming), and the production of harmful secondary metabolites (cyanotoxins), reduce the fitness of the [42,74] and mean they are a low quality diet for *Daphnia*. Our data show that the *δ*
^13^C isotopic signature of *D. magna* slowly decreases. After 24 h, the *δ*
^15^N values start to increase while the *δ*
^13^C values remain at the same level.

The increasing *δ*
^15^N levels suggest that the animals are in nutritional stress through starvation or as result of the low dietary quality and quantity [58,62,75,76,77]. Herein, we undertook a microscopic investigation of the guts of multiple individuals of *D. magna* and found that the individuals had ingested the offered food algae (gut observation). However, we did not measure the gut residence times or clearing rates so we were not able to exclude starvation-like symptoms due to the inefficient uptake of carbon through the thick cell walls or feeding deterrents. Most importantly, *S. elongatus* lacks sterols and is of poor food quality [42,48,49,71]. *T. variabilis* produces the C-18 PUFAs (similar to ALA and SDA) [42,48,49] but is deficient in long-chain PUFAs (e.g., C-20). Due to the absence of sterols and long-chain PUFAs, *Daphnia* individuals are likely to suffer from nutritional stress after 24 h, when the internal nutrient reserves are depleted. As a consequence, it is likely that they recycle their own somatic nitrogen [58,61,62,78], resulting in an increase in *δ*
^15^N values.

Previously, it has been suggested that the lack of sterols is the main reason for the low food quality of cyanobacteria for *Daphnia* and that the lack of C-20 PUFAs in cyanobacteria becomes relevant only when dietary sterol requirements are met by consuming eukaryotic food sources along with the cyanobacteria, potentially resulting in a co-limitation by sterols and C-20 PUFAs [71].

To summarize, the lipid composition of the primary producers can affect the fractionation rate of the consumers in two ways. Firstly, an increased lipid concentration in the primary producers can lead to an increased lipid content in the consumer tissues, depleted *δ*
^13^C values, and reductions in Δ^13^C values. Secondly, a PUFA-rich diet might lead not only towards an increased growth rate but also enhanced resource investment towards reproduction [42,79] and can potentially affect the isotope discrimination factors [80,81,82].

### 4.4. Toxic Unicellular Cyanobacteria

In contrast to the isotopic signatures of starving animals, *D. magna* samples fed with *M. aeruginosa* showed increases in *δ*
^13^C (1.57‰ ± 0.25‰) and *δ*
^15^N (1.17‰ ± 0.59‰) during exposure to the toxic unicellular cyanobacteria.

In addition to the mechanical interference through colony formation and the lack of essential nutrients, the cyanobacteria can affect the zooplankter fitness through the production of harmful secondary metabolites and therefore can influence their isotopic fractionation. The *M. aeruginosa* strain PCC 7806 is known to produce a variety of harmful metabolites such as microcystins, cyanopeptolins, and anabaenopeptins [37], which can lead to increasing mortality rates in *Daphnia*. In our experiment, the mortality rate of *D. magna* reached 100% after 96 h of exposure to *M. aeruginosa*. Due to their non-selective filtering process, *Daphnia* is unable to distinguish food particles in regard to nutritional quality. However, the perception of toxic phytoplankton leads to a complete inhibition of the filtering process [83], and individuals often display arrested feeding, at least for a while. In our study, it is likely that the putative inhibition of food intake induced a starvation-like state in the *Daphnia*, leading to an enrichment of the *δ*
^15^N and *δ*
^13^C values. *Microcystis* may also form aggregates, potentially leading to mechanical interference with the *Daphnia* filtration process. However, the microscopic investigation of the phytoplankton culture showed no formation of aggregates. Since we cannot rule out that aggregates were formed as a result of the feeding pressure, we examined the guts of *D. magna* microscopically and observed that the cells were ingested. This implies that the observed shifts in isotopic signatures were due to nutrient deficiencies or toxin production.

In the study by Brzeziński et al. [84], the *Daphnia magna* individuals became progressively enriched in *δ* ^15^N with the increasing concentration of a toxic chemical, while the stable carbon isotopes were not affected. Since all *Daphnia* individuals from all treatments were fed with food from the same source with the same isotopic signature, this ruled out diet-related effects on the stable isotopes. Compared to their results, the *D. magna* individuals in our experiment showed enrichment in both *δ* ^15^N and *δ* ^13^C while exposed to toxic cyanobacteria. The enrichment of stable nitrogen and carbon could occur due to reduced growth [81], alterations of metabolic pathways involving protein synthesis, and carbon turnover [85], as wells as the allocation of energy among detoxification processes [86,87], with an enhanced excretion rate of ^14^N for detoxification [88] and increased respiration.

### 4.5. Toxic Filamentous Cyanobacteria

Compared to the *M. aeruginosa* exposure, the *Daphnia* individuals fed *Planktothrix agardhii* and *P. rubescens* showed lower *δ* ^13^C values but stronger *δ* ^15^N enrichment, which indicated a starving process in *D. magna*. After 96 h, the *D. magna* individuals from the *P. rubescens* treatment showed the strongest *δ*
^15^N enrichment of the whole experiment, with an increase of 5.15‰ (± 0.85‰).

The used strains of *P. agardhii* as well as *P. rubescens* are not able to synthesize microcystins. Nevertheless, they produce harmful metabolites, such as anabaenopeptins [44,45,46]. Like microcystins, anabaenopeptins lead to the inhibition of *Daphnia* individuals’ swimming behavior [89] and alter their physiology [90]. Additionally, *Planktothrix* individuals can form long trichomes, which potentially interfere mechanically with the filtering process of *Daphnia* individuals and may reduce their grazing efficiency [91,92,93,94] and increase their metabolic rate [95]. However, *Daphnia* individuals are able to ingest particles of a wide size range, reaching from particles of less than 1 µm [96] up to mm-sized filamentous algae [97], including trichomes of *Planktothrix* [98]. Microscopic gut investigations revealed that the *D. magna* individuals ingested both strains of *Planktothrix* species. Therefore, mechanical interference and associated starvation due to lack of ingestion can be ruled out. The enriched isotope values indicative of starvation were likely caused by sterol limitations or the inhibitory effects of secondary metabolites. By implication, if the animals cannot express growth due to the lack of an essential nutrient or the inhibition of nutrient assimilation due to harmful metabolites, then the isotope signature ’falsely’ mirrors values that imply the lack of consumption of cyanobacteria. However, the shifts in isotopic signatures are due to nutrient deficiencies or the influence of toxins, despite the cyanobacteria ingestion. Finally, the production of harmful metabolites and the resulting toxic stress combined with the mechanical interference and the lack of sterols and long-chain PUFAs may explain the enrichment in *δ*
^15^N in *Daphnia* individuals fed with *P. rubescens* and *P. agardhii*.

## 5. Conclusions

In summary, in this study we parameterized and formulated stable isotope turnover rates and discrimination factors to estimate the timing of resource shifts of *Daphnia* individuals. The parameters generated in this paper for eight phytoplankton species and starvation provide a strong base for future stable isotope studies of this key primary consumer species. More generally, through model fitting and information based on the characterization of the phytoplankton species approach, this study provides an overview into the physiological reasoning underpinning stable isotope dynamics. Furthermore, through exploring the strengths and limitations of the different diets and the influence of their potential quality, quantity, and toxic stress on trophic isotope variations in *Daphnia* individuals, this study illustrates how temporal estimates of resource switching are affected, while the tissue turnover models and discrimination factors might also be influenced. It is evident that advances in stable isotope studies will be most effective when they can be supported by laboratory, theory, and field-based investigations.

## Figures and Tables

**Figure 1 biology-11-01816-f001:**
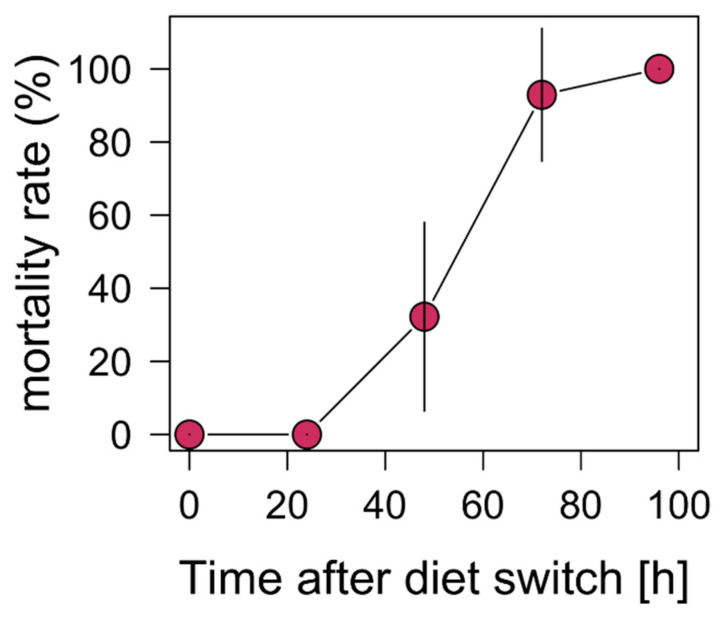
Mortality rate (%) for *D. magna* with the *M. aeruginosa* dietary treatment over a time period of 96 h following the dietary switch.

**Figure 2 biology-11-01816-f002:**
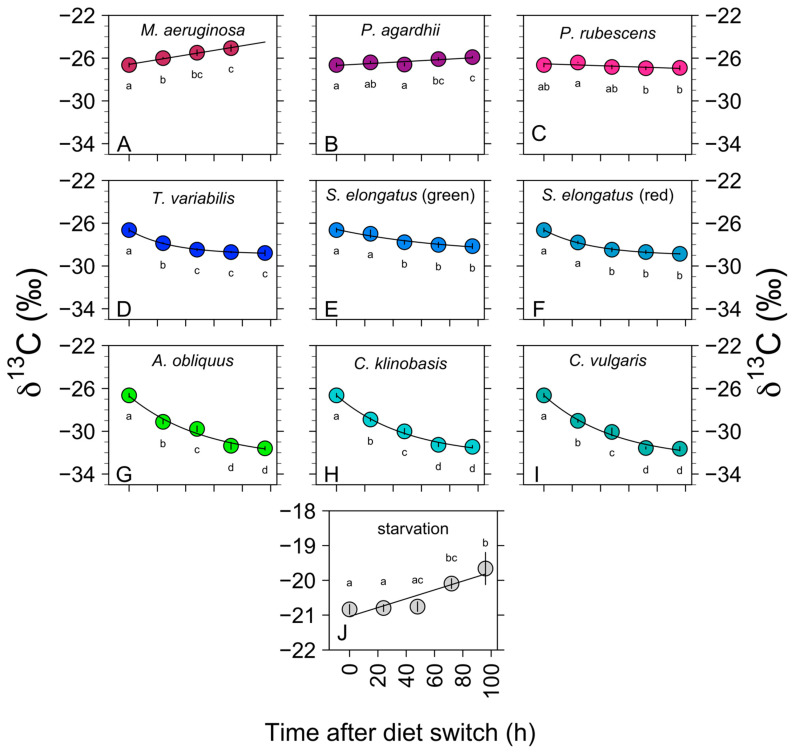
Carbon isotopic change in *D. magna* tissue shown as a function of time (hours) after changes to isotopically distinct captive diet types (or during starvation). The colored lines represent the signatures of the associated diet while the black lines represent the time-based exponential (**D**–**I**) or linear (**A**–**C**,**J**) model fits. The letters represent statistical differences for the *δ*
^13^C values of the different timepoints.

**Figure 3 biology-11-01816-f003:**
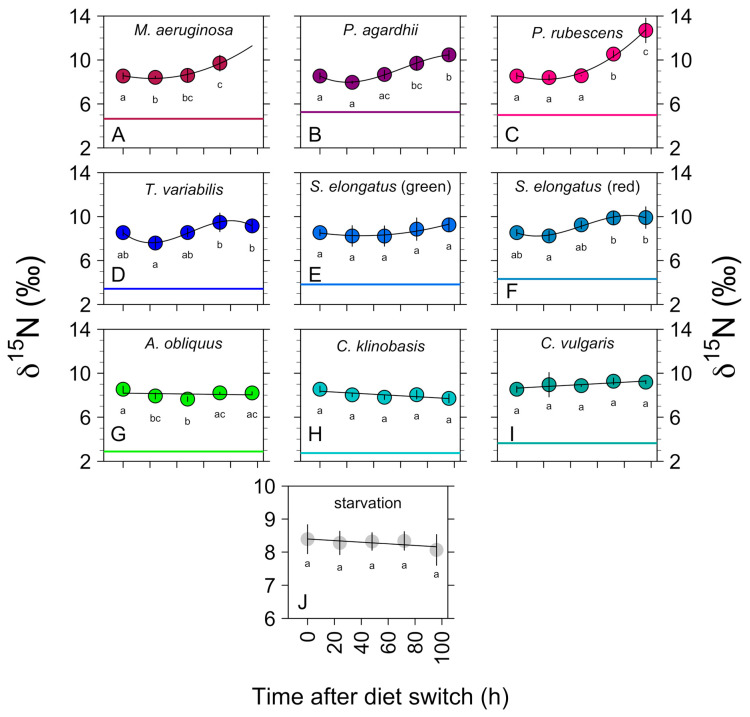
Nitrogen isotopic changes in *D. magna* tissue shown as a function of time (hours) after changes to the isotopically distinct captive diet types (or during starvation). The colored lines represent the signatures of the associated diets while the black lines represent the polynomial trendlines (**A**–**F**) or linear trendlines (**G**–**J**). The letters represent statistical differences for the *δ*
^15^N values of the different timepoints.

**Figure 4 biology-11-01816-f004:**
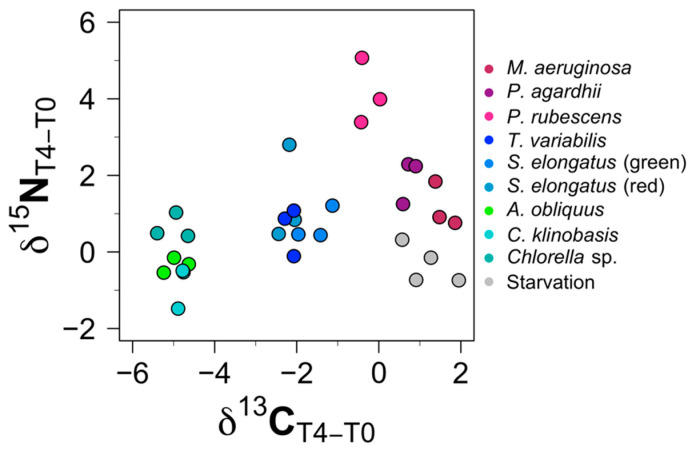
Changes in isotopic signatures of *D. magna* tissues over time (96–0 h after diet change) in the different dietary treatments.

**Figure 5 biology-11-01816-f005:**
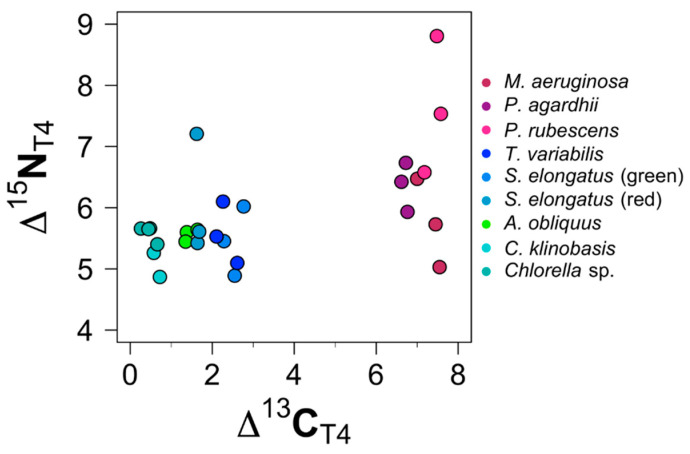
Discrimination factor (Δ) values for stable carbon and nitrogen isotopes 96 h (*M. aeruginosa* 72 h) after the dietary switch.

**Figure 6 biology-11-01816-f006:**
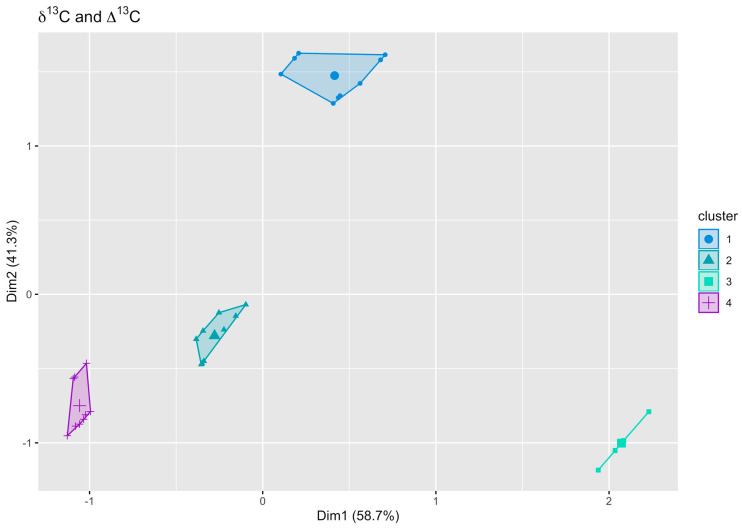
A cluster analysis (k-means) of the *δ*
^13^C and Δ^13^C values of *D. magna* fed with different dietary sources after 96 h (for *M. aeruginosa* after 72 h). Cluster 1 (blue) represents the cyanobacterial diet for *T. variabilis* and *S. elongatus* (both strains); cluster 2 (turquoise) represents *D. magna* fed on *M. aeruginosa, P. rubescens,* and *P. agardhii*; cluster 3 is shown in green and is formed by the chlorophytes *A. obliquus, C. klinobasis,* and *C. vulgaris;* and cluster 4 (violet) is formed by starving *D. magna* samples.

**Table 1 biology-11-01816-t001:** Phytoplankton species used as dietary sources for *Daphnia* and their respective origin, toxicity, polyunsaturated fatty acid (PUFA), and sterol provisioning information.

Phytoplankton	Origin	Toxic (+) Yes (−) No	PUFAs (+) Yes (−) No	Potentially Relevant PUFAs	Sterols (+) Yes (−) No	Potentially Relevant Phytosterols
*Microcystis aeruginosa*	PCC 7806	(+) [37,38,39,40,41]	<C 18 (+) [42] >C 18 (−) [42,43]	ALA [42,43] SDA [42]	(−) [42]	(−) [42]
*Planktothrix rubescens* No 91/1	MON (isolated Mondsee 2001, Kurmayer)	(+) [41,44,45,46]	no info.	no info.	no info.	no info.
*Planktothrix agardhii* No 829	MON (isolated Russland 2008, Kurmayer)	(+) [41,44]	no info.	no info.	no info.	no info.
*Trichormus variabilis* P9	ATCC 29413	(−) [47]	<C 18 (+) [42,48] >C 18 (−) [42,49]	ALA [42,48,49] SDA [42,49]	(−) [42]	(−) [42]
*Synechococcus elongatus* Bo 8801 (green)	KON 76 (isolated Lake Constance)	no info.	(−) [42,48,49] *	(−) [42,48,49] *	(−) [42,50] *	(−) [42,50] *
*Synechococcus elongatus* Bo 8809 (red)	KON 77 (isolated Lake Constance)	no info.	(−) [42,48,49] *	(−) [42,48,49] *	(−) [42,50] *	(−) [42,50] *
*Acutodesmus obliquus*	SAG 276-3a	no info.	<C 18 (+) [42,48] >C 18 (−) [49]	ALA [48] SDA [48]	(+) [51,52]	chondrillasterol [51] fungisterol [51] 22-dihydrochondrillasterol [51]
*Chlamydomonas klinobasis*	KON 56 (isolated lake constance)	no info.	>C 18 (−) [53]	ALA [53]	(+) [51,54]	ergosterol [51] 7-dehydroporiferasterol [51]
*Chlorella vulgaris*	KON 65 (isolated lake constance)	no info.	<C 18 (+) [55] >C 18 (+) [55] *	ALA [55] * EPA [55] * DHA [55] *	(+) [51]	ergosterol [51] * fungisterol [51] *

PUFAs = polyunsaturated fatty acids; * = not the same strain; no info. = no information known.

**Table 2 biology-11-01816-t002:** Mean (±SE) stable isotope values and C/N ratio of each phytoplankton species used as a dietary source for the diet switch experimental setup.

Diet	*δ* ^13^C	*δ* ^15^N	C/N Ratio
*A. obliquus* (Cyano-Medium)	−25.59 ± 0.09	3.10 ± 0.01	2.86 ± 0.16
*M. aeruginosa*	−32.28 ± 0.18	4.65 ± 0.85	3.43 ± 1.22
*P. agardhii*	−32.80 ± 0.17	5.26 ± 1.35	3.10 ± 0.65
*P. rubescens*	−34.35 ± 0.15	4.99 ± 0.36	2.74 ± 0.20
*T. variabilis*	−31.44 ± 0.22	3.43 ± 0.17	2.50 ± 0.06
*S. elongatus* (green)	−30.53 ± 0.16	3.83 ± 0.12	2.75 ± 0.21
*S. elongatus* (red)	−30.40 ± 0.08	4.32 ± 0.35	2.56 ± 0.21
*A. obliquus*	−32.99 ± 0.18	2.80 ± 0.23	2.19 ± 0.09
*C. klinobasis*	−32.05 ± 0.10	2.75 ± 0.48	2.63 ± 0.15
*C. vulgaris*	−32.08 ± 0.04	3.63 ± 0.12	2.89 ± 0.21

**Table 3 biology-11-01816-t003:** A comparison of the *δ*
^13^C values for the diet and *D. magna* at the end of the experimental timeframe (96 h, and for *M. aeruginosa* 72 h) (******* = < 0.001; ****** = <0.01 & > 0.001; ***** = < 0.05 and > 0.01).

Diet	t	df	Significance Level	Statistical Test	Significance Level
*M. aeruginosa*	48.336	2	***	Paired t-test	***
*P. agardhii*	−50.803	2	***		
*P. rubescens*	−40.07	2	***		
*S. elongatus* (green)	−10.877	2	**	Paired Wilcoxon test	**
*S. elongatus* (red)	−20.385	2	**		
*T. variabilis*	−11.746	2	**		
*A. obliquus*	−13.183	2	**	Paired Wilcoxon test	**
*C. klinobasis*	−8.8552	2	*		
*C. vulgaris*	−3.6104	2	n.s.		

**Table 4 biology-11-01816-t004:** Parameter estimates and standard errors of the linear and exponential decay function fitted to the *δ*
^13^C values of *D. magna* samples fed on different dietary sources over an experimental time period of 96 h and a comparison with Akaike’s Information criterion corrected for the small sample size (ΔAICc). Here, *δ*_0_ it the initial isotopic ratio of the experiment, *δ_eq_* is the asymptote (plateau) of the isotopic ratio, and *λ* is the incorporation rate of *δ*
^13^C. The models were fitted using a time-based model for each dietary source (*n* = 3) and for pooled data for the three diets (*n* = 9), respectively. E = exponential decay model; L = linear decay model.

Time (h)	Diet	*δ* _0_	*δ_eq_*	log*_λ_*	*λ*	*AIC*	∆*AIC*	Half-Life *δ* ^13^C	
96	*T. variabilis*	−26.64 ± 0.03	−28.89 ± 0.03	−3.39 ± 0.05	0.034	−18.28 (E)	27.23	20.46	27.63
96	*S. elongatus* (green)	−26.58 ± 0.18	−28.92 ± 1.20	−4.39 ± 0.87	0.013	0.95 (E)	5.84	55.69	
96	*S. elongatus* (red)	−26.63 ± 0.05	−29.01 ± 0.07	−3.54 ± 0.08	0.029	−13.18 (E)	21.24	23.78	
96	*A. obliquus*	−26.70 ± 0.41	−32.64 ± 1.26	−3.99 ± 0.45	0.019	8.78 (E)	4.6	37.28	35.52
96	*C. klinobasis*	−26.65 ± 0.22	−32.38 ± 0.60	−3.91 ± 0.23	0.020	2.67 (E)	10.52	34.48	
96	*C. vulgaris*	−26.65 ± 0.33	−32.68 ± 0.91	−3.92 ± 0.34	0.020	6.60 (E)	7.34	35.02	
72	*M. aeruginosa*					−6.05 (L)			-
96	*P. agardhii*					11.05 (L)			
96	*P. rubescens*					17.74 (L)			

## Data Availability

Not applicable.

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
