# Peer review of "Toxicity and Starvation Induce Major Trophic Isotope Variation in Daphnia Individuals: A Diet Switch Experiment Using Eight Phytoplankton Species of Differing Nutritional Quality"

_biology, 2022, doi:10.3390/biology11121816_

Round 1

Reviewer 1 Report

Thank you for the interesting and relevant study! Quantitative values for planktonic parameters are certainly necessary for reference in further studies. 

I do not have many comments on the manuscript, however, some aspects still need to be improved. 

1) In the "Introduction" paragraph, starting at the line 61, the second sentence (lines 62-65) needs some rewording. It is hard to understand what exactly is happening. I got the idea but it should be expressed more clearly. In the next sentence (lines 65-67) metabolism is named as an example of anabolic and catabolic processes. Well, anabolic and catabolic processes themselves constitute the metabolism, so another example should be provided.

In the next paragraph (lines 69-83) phytoplankton community changes due to climate variation are described. Still, zooplankton planktivores do not necessarily feed on all cyanobacteria. I'd suggest to include also this aspect for clarity.

In "Discussion", Lines 449-451 it is mentioned that microscopic examination of Daphnia guts was performed. At first, it look like two sentences have not been successfully combined here ("If" in the middle of the sentence) and secondly, what was the result of this examination? Did you see the algae in the guts? Please clarify here. 

2) Headlines of chapters

Here more consistency is necessary and better wording should be sought, please see the list below.

a) Line 179 - is it a title for a sub-chapter? If yes, probably it should be formatted accordingly.

b) 2.4., Line 194 - Determination of the best model fit?

c) 3.1., Line 227 - it is not just phytoplankton where daphnids were kept. Please use more precise wording

d) 3.1.2. , Line 245 - capital letter missing 

e) in Discussion I'd suggest to rethink the idea call the clusters of consumers according to their diet objects. 

3) The quality of the text

It would help if Daphnia is used correctly in Latin, and daphnids in English. Please check the consistency of the use!

Reviewer 2 Report

Carbon and nitrogen stable isotope analysis are useful tools to study trophic interactions between organisms in freshwater environments also in relation to pollutants. Nevertheless, physiological mechanisms that underpin the isotopic fingerprint of organisms are very complex and not completely understood. In this contest, the topic discuss in the present manuscript is very interesting and improve the scientific knowledge, but in my opinion the paper cannot be published in its present form, my advice is: major revision.

In the introduction paragraph the authors pointed out the influence of climate change on freshwater environments leading to favour Cyanobacteria. In this context, it is clear the choice to test Cyanobacteria species, but why do you selected filamentous species which are known to be not very edible for Daphnia? Likely, Daphnia specimens are able to eat only small fragments derived from broken algal filaments. Studies on trophic webs are based on the assumptions that isotopic carbon fingerprint is quite conservative along the food chain (according to DeNiro & Epstein (1978), a common carbon source is attributed when fractionation: F= δ13Cpredator - δ13Cprey ≤ 0.8 ‰ ±1.1 ‰ S.D.), while there is an isotopic nitrogen enrichment between preys and predators. Results reported in the MS highlighted clearly a great difference of d13C values of daphnids from algae in Planktotrix treatments. In my opinion, these results indicate that daphnids are not feeding. The same consideration can be done for Microcystis treatment, likely because of the production of toxic compounds or because this algal species forms big aggregates which are not easily eaten by daphnids. While the shift of the d13C values of daphniids towards the d13C‰ of algal species indicates an active feeding in treatsments with A. obliquus, C. klinobasis and C. vulgaris.

Other specific comments and editing changes are reported on pdf file.

Round 2

Reviewer 2 Report

The authors have addressed to all questions pointed out in the review, improving the quality of the paper.

My advice is to accept the manuscript in its present form.